# Spatiotemporal Evolution of Evapotranspiration in China after 1998

**Qi Guo****, Jiening Liang, Xianjie Cao, Zhida Zhang and Lei Zhang ***

Key Laboratory of Semi-Arid Climate Changes with the Ministry of Education, College of Atmospheric Sciences, Lanzhou University, Lanzhou 730000, China; guoq17@lzu.edu.cn (Q.G.); liangjn@lzu.edu.cn (J.L.); caoxj@lzu.edu.cn (X.C.); zhangzhd14@lzu.edu.cn (Z.Z.)
* Correspondence: zhanglei@lzu.edu.cn

**Abstract:** Changes in water circulation and uneven distributions of water resources caused by global warming are prominent problems facing the world at present. It is important to understand the influencing factors, and evapotranspiration (ET) is a key parameter for measuring the water cycle. However, understanding of spatiotemporal changes in actual evapotranspiration and its mechanism is still limited by a lack of long-term and large-scale in situ datasets. Here, the evolution of evapotranspiration in typical East Asian monsoon areas in China from 1989 to 2005 was analyzed with global land ET synthesis products. Evapotranspiration in China showed evident interdecadal variations around 1998; it decreased before 1998 and subsequently increased, which is inversely related to global ET changes. We further divided China into water-control and energy-control regions to discuss the factors influencing ET changes in each region. The interdecadal variations in increasing ET after 1998 in China were dominated by increasing potential evaporation in the energy-control region. An analysis using the empirical orthogonal function (EOF) method found that this occurred because ET is mainly manifested as decadal changes controlled by climate warming in the energy-control region and as interannual variations in the water-control region. The different feedbacks of ET on climate change in the two regions were also reflected in the difference in energy partition. The change in the Bowen rate (BR) did not increase climatic differences between energy- and water-control zones, but increases in the BR in arid summers significantly affected local weather and climate.

**Keywords:** evapotranspiration; potential evaporation; water balance; EOF; energy distribution

## 1. Introduction

Most of the precipitation that falls to the ground returns to the atmosphere in the form of evapotranspiration except for runoff and local storage, and evapotranspiration accounts for 60% on land [1]. Changing the water cycle and evapotranspiration will affect regional ecosystems and climates under global warming and cause feedbacks, including acidification and heat waves [2–7]. The spatiotemporal distributions and trends in evapotranspiration appear to be particularly important [8–10].

The Intergovernmental Panel on Climate Change [11,12] noted that the global average land temperature has increased by 0.74–0.85 °C in the past 100 years, which would directly affect the water content and cycle in the atmosphere, including global precipitation and evapotranspiration. As a critical variable connecting the water balance and the energy balance, ET (evapotranspiration) can be an excellent indicator of water cycle changes and even climate changes. For every 1 °C increase in temperature, the atmospheric water content increases by 7% according to the Clausius–Clapeyron equation [13,14]. However, evaporation does not show the expected increasing trend under global warming. Many studies have shown that pan ET and potential evaporation (PET) have decreased

dramatically on a global scale over the past few decades [15–17]. Temperature and evaporation show the opposite trends under global warming, which is called the "evaporation paradox". This is because the relationship between actual ET and pan ET or PET is spatially complex. Research in recent years has noted that the relationship between actual ET and PET is complementary to water control and proportional to energy control [18]. That is, actual ET is restricted by soil moisture in arid areas, while in areas with sufficient water supply, it is mainly controlled by atmospheric evaporation demand [19–21]. PET represents the atmospheric evaporation demand, which is affected by numerous weather conditions. Climate change can affect PET through a combination of temperature, net radiation, wind speed, and vapor pressure deficit. This makes the actual ET response to climate change extremely complex in space and time.

Previous studies focused on small areas and did not consider the differences in ET evolution under different attributes. Azorin-Monlina et al. found that temperature and relative humidity are the main contributors to evaporation [22]. Wang et al. and Liu et al. noted that wind speed reduction is the most important contributor in China [23,24]. Sun et al. also found that the factors affecting evapotranspiration are different in different subregions [25]. Hence, to study the factors influencing ET, we need to classify these factors according to the attributes of the underlying surfaces. In addition, evapotranspiration is an important source of water vapor for local precipitation. Correspondingly, against the background of global warming, the variations in global precipitation are not large, and they are also mainly manifested as large regional differences. Such uneven changes in evapotranspiration and precipitation result in an uneven distribution of freshwater resources and the frequent occurrence of drought and flood events, which seriously affect people's lives and the safety of property. China is located in the typical East Asian monsoon area that is sensitive to climate change, so the evolution of evapotranspiration in China can be discussed in terms of different regions to reflect the feedback of the water cycle to climate change. Clarifying the evolutionary differences and contributing factors of evapotranspiration in different regions is the basis of coping with regional climate change.

At present, the large-scale accurate observation of evapotranspiration mainly comes from the latent heat data obtained by the eddy correlation method at FLUXNET sites. However, flux site distribution is very sparse compared with conventional meteorological observation stations, and sites with long-term reliable observations are even rarer. Other methods to obtain ET data mainly include the following: diagnostics according to surface water balance and atmospheric water balance, satellite-based retrieval algorithms and empirical models, and output from land surface models. Different methods have their own applicabilities, advantages and disadvantages, and different ET trends have been estimated around the world [26–28]. China flux sites and long-term effective observational data are particularly lacking, and studies on the spatiotemporal changes in ET in China usually replace evaporation with pan ET or PET from the computations based on conventional meteorological data, which cannot represent the evolution of actual ET because surface properties are not considered [29–32].

Recent years have seen improvements in global data assimilation and reanalysis systems, and the understanding of hydrological circulation has notably improved. Su et al. used five reanalysis datasets to study the evolution and influential factors of evapotranspiration in China and found that ET increased in northwestern and southern China under global warming [33]. However, no definite interdecadal changes in evaporation were observed. Wang et al. noted that flash droughts are three times more frequent in China than they had been previously, which is closely related to increasing ET in China [34]. Peng et al. also found that the increase in local evaporation over the past few decades is the main reason for the increase in precipitation in western China [35]. An increasing number of studies are focusing on the changes in Chinese evapotranspiration and their influence on weather and climate. At present, the uncertainty among different reanalysis datasets in describing ET changes remains unknown. These studies often consider one or two reanalysis datasets as the optimal observation, and the use of actual and potential evaporation is ambiguous. The decadal evolution of evaporation as an important component of the water cycle in China, a key region of the East Asian monsoon, is unclear.

Understanding and exploring the interannual and interdecadal variabilities in ET is an essential part of assessing climate variability and possible changes. To correctly assess the impact of climate change on water resources and regional climate differences in China, new multiyear synthesis products that emerged from FLUXNET, a diagnostic estimation and satellite retrieval dataset, are used, which can represent actual global evapotranspiration. We aim to answer the following questions: (1) How have actual ET and PET evolved in the typical East Asian monsoon region of China under global warming? (2) In regions with different attributes, what are the dominant meteorological elements that affect interannual and interdecadal evaporation? To some extent, we hope to provide a reference for the feedback of actual evaporation in East Asia to enhance our understanding of hydrological processes in different climate regions with the context of global warming.

## 2. Data and Methods

### 2.1. Study Area and Data

East Asia is a vast territory in a monsoon zone sensitive to climate change, with elevations ranging from less than 10 m in the east to more than 5000 m in the west and precipitation decreasing from the southeast to the northwest regions of China. Mainland China was selected as the study area. The complex and varied topography and climate patterns lead to highly variable land–atmosphere interactions in this region, which provide great natural conditions for the study of evapotranspiration. Conventional meteorological data such as temperature, precipitation, wind speed, net radiation, and relative humidity for this study are provided separately by datasets that perform well for each variable. For temperature, precipitation, and potential evapotranspiration, data from the Climatic Research Unit (CRU) at the University of East Anglia are used. CRU is recognized as providing a high-quality dataset, and its temperature and precipitation data are highly reliable [36]. Regional comparisons with other published datasets show that CRU temperatures agree closely with the University of Delaware (UDEL) dataset, and the correlation is 0.94 in East Asia. Soil moisture and relative humidity are provided by NCEP-DOE Reanalysis 2 (NCEP-R2) from the National Centers for Environmental Prediction [37]. NCEP-R2 improves the physical process parameterization scheme based on the previous version and provides reliable data for studying soil moisture. Net radiation, wind speed, sensible heat, and latent heat flux are determined using ERA-Interim data from the European Centre for Medium-Range Weather Forecasts (ECMWF). ECMWF and NCEP reanalysis data are widely used [38,39], and the application of these reanalysis data is highly convenient. All data are based on the monthly averages for 1989–2005 in China and are interpolated into a $1 \times 1°$ grid to match the time and spatial scales of the evapotranspiration dataset. In addition, the sensible and latent heat data of the FLUXNET site in China are downloaded for comparison and verification. The details of the data are shown in the Table 1.

**Table 1.** Reanalysis datasets and variables used.

| Datasets | Variables | Time Scale | Resolution |
|---|---|---|---|
| LandFlux-EVAL | Evapotranspiration | 1989–2005 (Monthly) | $1 \times 1°$ |
| FLUXNET | Sensible heat flux, Latent heat flux | 2003–2005 (Monthly) | Site |
| CRU | Temperature, Precipitation, Potential evaporation | 1989–2005 (Monthly) | $0.5 \times 0.5°$ |
| NCEP/DOE Reanalysis 2 | Soil moisture, Relative humidity | 1989–2005 (Monthly) | $1 \times 1°$ |
| ERA-Interim | Net radiation, Wind speed, Sensible heat flux, Latent heat flux | 1989–2005 (Monthly) | $1 \times 1°$ |

The LandFlux-EVAL datasets are used for actual evapotranspiration; LandFlux-EVAL products are new multiyear merged synthesis data based on the analyses of land ET datasets that are available to us [40,41]. LandFlux-EVAL are the first benchmark synthesis products for monthly global land ET estimates. The merged synthesis products are based on a total of 40 distinct datasets. In individual datasets, ET is derived from satellite and/or in situ observations (diagnostic datasets) or calculated via

land surface models (LSMs) driven by observation-based forcing or output from atmospheric reanalyses. The output statistics for each of the merged synthesis products are the mean, median, 25th percentile, 75th percentile, interquartile range, standard deviation, and minimum and maximum values of the ensemble of underlying datasets. All products are available at monthly and yearly temporal resolutions and as multiyear statistics. The study period in this paper is limited to 2005 because LandFlux-EVAL products are available only for 1989–2005. We also downloaded other available reanalyses (MERRA2, NCEP2, and ERA5) to provide additional information on the variability in ET after 2005, and the relevant content is included in the Supplementary File.

The applicability of this global dataset has been verified [40–43], and we need to verify its reliability in China. A scatter correlation analysis is implemented using latent heat data from the FLUXNET China stations with LandFlux-EVAL, as shown in Figure 1. The FLUXNET sites in China are sparse, and the data period is short. Here, flux data for 2003–2005 are used from several sites, including Changbai Mountain (cha), Dinghu Mountain (din), Qianyanzhou (qia), Yucheng, Haibei (ham, ha2), Inner Mongolia and Dangxiong (dan) stations. The resolution of LandFlux-EVAL is monthly, averaged on a 1° uniform grid, and we linearly interpolated it to the corresponding latitude and longitude at each FLUXNET site for comparison and verification. The figure shows that the correlation between the two datasets is good at each station, and the correlation coefficient is greater than 0.9 for the cha, ha2, ham, and dan stations. The correlation is best in cha and poorest in qia, where the correlation coefficient is only 0.8. The LandFlux-EVAL dataset for the Dinghu Mountain site shows a larger ET value than the FLUXNET observation, while it is smaller at the Dangxiong and Heibei sites. Overall, all points in the figure are close to the 1:1 line, and LandFlux-EVAL can be considered applicable for the actual ET in China.

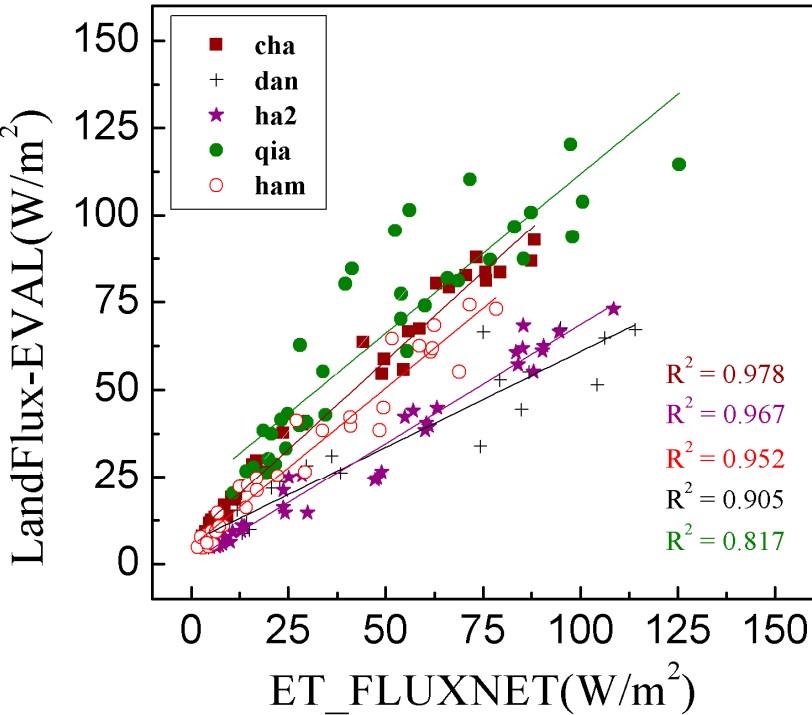

**Figure 1.** The scatter correlation of ET (evapotranspiration) monthly averages from FLUXNET with LandFlux-EVAL at 5 sites in China for 2003–2005. Sites are represented by different colors and shapes. The straight line is the linear fit for each site. R2 is the correlation coefficient.

### 2.2. Methods

The Mann–Kendall (MK) trend test and the Theil–Sen trend estimate technique [44,45] are utilized in this study to evaluate the significance of the climatic series trends. The rank-based nonparametric

Mann–Kendall test can test trends of a time series without requiring normality or linearity [46]. The MK test is widely used for identifying trends in hydrological and meteorological data. The MK test first calculates an *S* statistic:

$$S = \sum_{k=1}^{n-1} \sum_{j=k+1}^{n} \text{sgn}(x_j - x_k),$$

(1)

where $x_j$ are values in $j$th years, $n$ is the number of observations, and the function $sgn(x)$ gives the sign of the variable $x$. The statistic $S$ is approximately normally distributed under the null hypothesis. If $n$ is greater than 10, the variance of the $S$ statistic is computed by:

$$Var(S) = \frac{1}{18}[n(n-1)(2n+5) - \sum_{p=1}^{q} t_p(t_p-1)(2t_p+5)],$$

(2)

where $q$ is the number of tied groups and $t_p$ is the number of data values in the $p$th group. The standardized test statistic $Z$ is computed as follows:

$$Z = \begin{cases} \frac{S-1}{\sqrt{Var(S)}} & if \ s > 0 \\ 0 & if \ s = 0 \\ \frac{S+1}{\sqrt{Var(S)}} & if \ s < 0 \end{cases}.$$

(3)

The Theil–Sen test yields a robust estimate of the linear trend. It can be significantly more accurate than simple linear regression for skewed and heteroskedastic data and competes well against non-robust least squares even for normally distributed data in terms of statistical power [47]. The Theil–Sen estimator is calculated by:

$$\beta = Median(\frac{x_j - x_k}{j - k}), \ \forall \ 1 < k < j < n,$$

(4)

where $\beta$ is the median overall combination of recorded pairs for all data. A positive $\beta$ value indicates an upward trend, and a negative $\beta$ value indicates a downward trend.

The empirical orthogonal function (EOF) is applied in the energy-control region and the water-control region to analyze the evolution forms of ET and the specific contributions in each region. The EOF is a common modal decomposition method in meteorology that can extract the dominant signals of meteorological fields well and reflect the variation characteristics of temporal and spatial distributions of such fields [48]. The main spatial distribution structure of the variable field is effectively separated; that is, based on preserving the information in the original variable field as much as possible, the influence of various meteorological elements is concentrated on the first few main components that explain the large variance, which is mathematically a kind of dimensionality reduction processing method [49]. Fundamentally, the EOF reduces the dimensions of multivariate data ($x_{i,k}$) by creating new variables ($y_{i,k}$) that are linear functions of the original variables:

$$\begin{bmatrix} y_{i,1} = p_{11}x_{i,1} + p_{12}x_{i,2} + \cdots + p_{1k}x_{i,k} \\ y_{i,2} = p_{21}x_{i,1} + p_{22}x_{i,2} + \cdots + p_{2k}x_{i,k} \\ \cdots \\ y_{i,k} = p_{k1}x_{i,1} + p_{k2}x_{i,2} + \cdots + p_{kk}x_{i,k} \end{bmatrix},$$

(5)

where $k$ is the number of variables and $i$ is the time period. The coefficients of the linear combinations are called loadings (i.e., eigenvectors), and they provide the weights of the original variables in principal components (PCs).

The aridity index (*AI*) is used to rank the grid points in China according to the degree of drought and wetness. The ratio of annual precipitation (*P*) to potential evapotranspiration (*PET*) is defined as the *AI* [50,51], which keeps the two units the same (e.g., mm):

$$AI = \frac{P}{PET} \tag{6}$$

The Bowen ratio (BR) is defined as the ratio between the sensible heat flux ($H_s$) and the latent heat flux (*LE*) [52,53]. We use a modified calculation to ensure that the *BR* is always negative when the local sensible heat flux is negative (i.e., downward) regardless of the sign of the latent heat flux [54]. The *BR* is calculated with Equation (6) using annual mean data at each grid and then averaged regionally; the units of *Hs* and *LE* are W/m$^2$:

$$BR = \left| \frac{H_s}{LE} \right| \times \text{sgn}(H_s). \tag{7}$$

We use the Budyko curve [55] to verify whether the division of dry and humid regions is reasonable. The Budyko curve describes the long-term average relationship between the aridity index (*PET/P*) and the evaporation index (*ET/P*). We can estimate the wet and dry properties of an area by the positions of monthly precipitation, evaporation, and potential evaporation in the Budyko curve. All the units are expressed in W/m$^2$ for comparison.

## 3. Results and Discussion

### 3.1. Spatial-Temporal Distribution and Trend of Evapotranspiration in China

First, the basic distribution and evolution of evapotranspiration in China are analyzed. We estimate a mean annual nationwide land ET value from 1989 to 2005 of 1.188 mm/day, with the spatial distribution shown in Figure 2a. The long-term annual mean ET across China shows a clear southeast-northwest gradient, and ET gradually decreases from the southeastern coast to the northwestern inland region of China. In the southeastern coastal area, the value is the largest at approximately 4.0 mm/day, and in the northwest, especially in Xinjiang Province, the value is the smallest. Figure 2b depicts the annual mean MK trend of ET from 1989 to 2005. Except for the northeastern part of Inner Mongolia and part of the Qinghai-Tibetan Plateau, the trend distribution is positive nationwide. The national annual ET increased on average by 0.665 mm/day/year, which is consistent with the results of Su et al. [33].

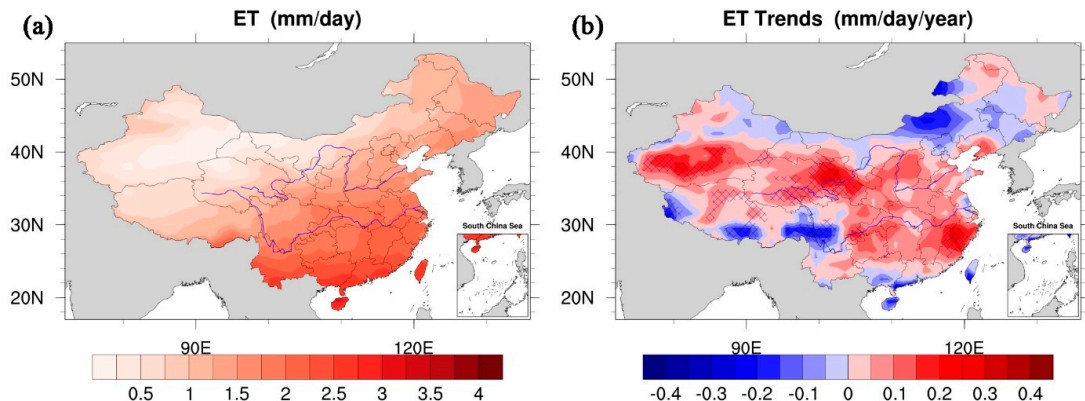

**Figure 2.** Distributions of the mean annual land ET (**a**) and the ET MK (Mann–Kendall) trend (**b**) in China from 1989 to 2005. Blue is negative, and red is positive. The grid lines indicate that the 95% significance test is passed.

The ET MK trend distributions of the multiyear averages in winter and summer after 1998 are shown in Figure 3a,b, respectively. The winter and summer distributions of the ET trend are nearly opposite, and the overall summer trend is stronger than that in winter, especially south of the Yangtze

River, which shows a significant increase in summer and a decrease in winter. The Qinghai-Tibetan Plateau and the southern part of Xinjiang have positive trends in winter and summer; the summer trend is stronger, and a positive trend covers nearly the entire Qinghai-Tibetan Plateau, Xinjiang, Qinghai, and southeastern Gansu Provinces. In Central China north of the Yangtze River, the trends are also opposite in winter and summer, with ET increasing in winter and decreasing significantly in summer. Overall, there are obvious seasonal differences in ET after 1998, and most cities experience a significant increase during summer.

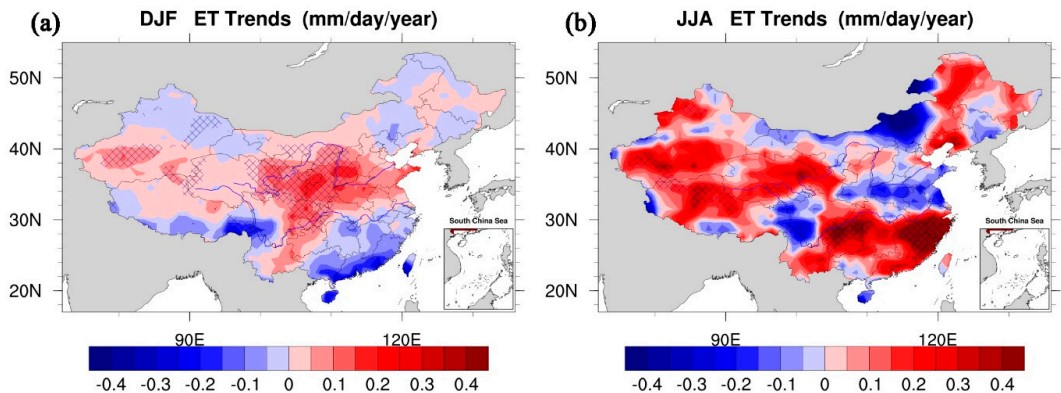

**Figure 3.** ET MK trend distributions of multiyear averages in winter (**a**) and summer (**b**) after 1998. Blue is negative, and red is positive. The grid lines indicate that the 95% significance test is passed.

*3.2. Regional Division in China*

Changes in meteorological factors such as air temperature, water vapor pressure, wind speed, and net radiation will affect evapotranspiration under climate change, and the interplay of factors complicates evapotranspiration [56–58]. The concepts of water-limited and energy-limited evaporation have long been used to understand the role of evaporation [15,59]. Therefore, when analyzing the causes of ET transformation, we can first classify ET changes into two categories: limited by water and limited by energy. Then, we can analyze the effects of meteorological factors in each region. Here, we use the criterion of whether the local water supply or precipitation can meet the atmospheric evaporation demand to divide China into two parts, which are mainly restricted by soil moisture and by atmospheric evaporation demand, as shown in Figure 4. That is, the area of significant positive correlation between ET and temperature is classified as restricted by atmospheric evaporation demand, and the area of significant positive correlation between ET and precipitation is classified as restricted by soil moisture [2]. Regions that are not significantly correlated with temperature and precipitation are divided into transitional zones. Northwest, north, and northeast China, which are blue in the figure, are mainly restricted by soil moisture (water-control region). Areas south of the Yellow River in Central China and South China, which are red in the figure, are mainly limited by atmospheric evaporation demand (energy-control region). Eastern Tibet and Heilongjiang Province are intermediate transitional areas that are represented by white in the figure.

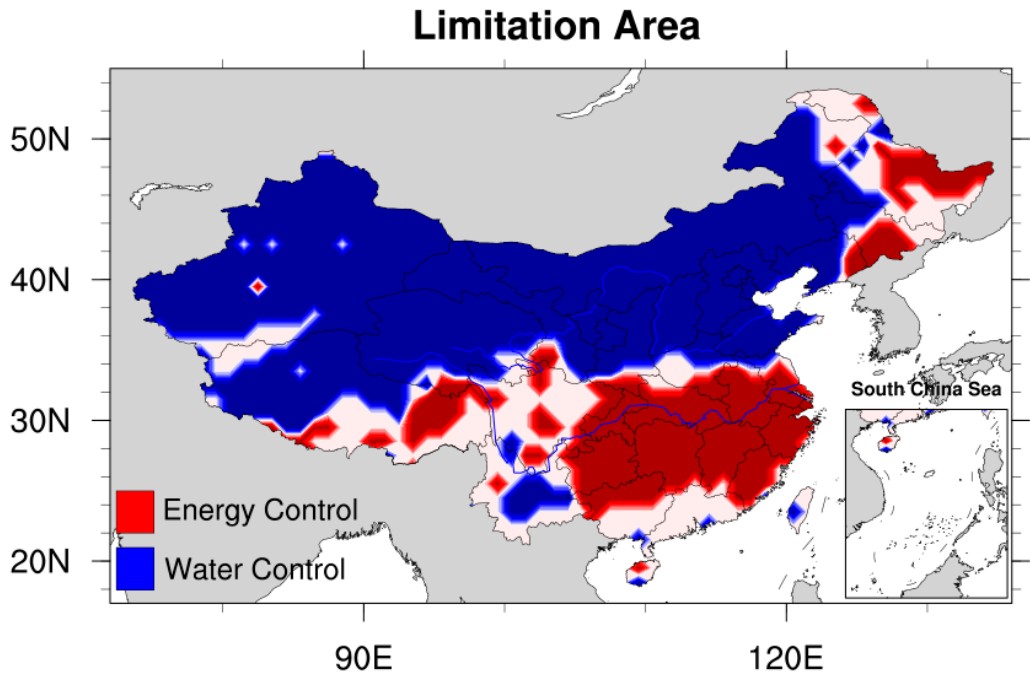

**Figure 4.** The limits for areas of energy control (in red) and water control (in blue) from calculating the correlation between annual ET and temperature for 1989–2005. The area of significant positive correlation between ET and temperature is classified as restricted by atmospheric evaporation demand, and the area of significant positive correlation between ET and precipitation is classified as restricted by soil moisture.

Subsequently, the Budyko curve was used to verify that the partition above is reasonable. The x-coordinate is the aridity index, and the higher the value is, the drier the region. The y-coordinate is the evaporation index, and a value closer to 1 indicates a higher evaporation ratio. The horizontal line in the figure indicates that water is limited, and all precipitation is taken up by ET. The slash indicates that the energy is limited, and all PET becomes ET. All the points for the two areas previously divided are plotted on the Budyko curve diagram, as shown in Figure 5. The points from the energy-control region show lower aridity index values, and the scatter is lower to the left. The span of the evaporation index is large. The points from the water-control region show a higher aridity index, and the scatter is higher to the right; the span of the evaporation index is small. These results prove that the two regions we identified previously, which are limited by soil moisture and atmospheric evaporation potential, are feasible. The division is not ideal, but on a large scale, the two regions can be clearly distinguished. The two regions we marked in Figure 4 are also consistent with those reported by Han et al. [60], who identified dry and wet grid cells based on observed precipitation data from the China Meteorological Administration from 1961 to 2014. Thus, we can study the relationship between ET and meteorological factors in the two regions with different restrictions.

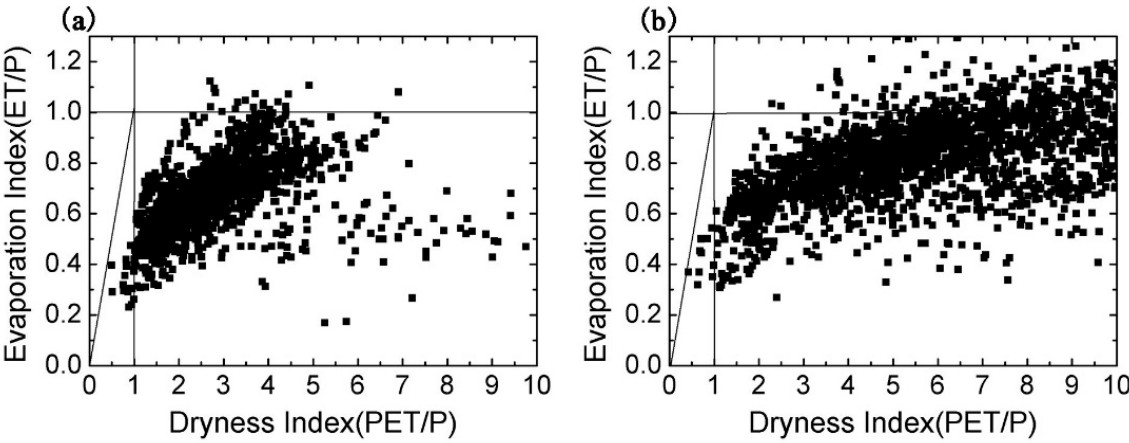

**Figure 5.** Scatters from the energy-control (**a**) and water-control (**b**) regions on the Budyko curve. The closer the scatter slope is to 1:1, the more PET (potential evaporation) is converted to evaporation, and the wetter the climate is.

### 3.3. Evolution and Attribution of ET in Different Regions of China

Figure 6 shows the national mean evolution of the ET anomaly from 1989 to 2005 in China, which is estimated using the annual average minus the climate average for 1989–2005. There are significant interdecadal changes in ET, suggesting that land ET decreased from the early 1980s to the late 1990s. This trend of decreasing land ET disappeared after the last large El Niño event in 1998, and the trend of the median land ET anomalies derived from these models became positive during 1998–2005. The evolution patterns of mean ET anomalies in NCEP2 and MERRA2 are consistent with that in LandFlux-EVAL before 2005 (Figure S1). ET continued to increase from 1998 to 2016, suggesting an inflection point for ET in 1998. This is opposite to the global ET trend during the same period, which increased before 1998 and decreased afterward. Distinguishing the land ET response due to atmospheric demand from that due to terrestrial moisture supply limitations is a classic ecohydrological problem. Global ET began to decrease after 1998, mainly due to limitations in soil moisture in the Southern Hemisphere [61]. Similarly, it is important to understand the unique evolution of ET in China around 1998.

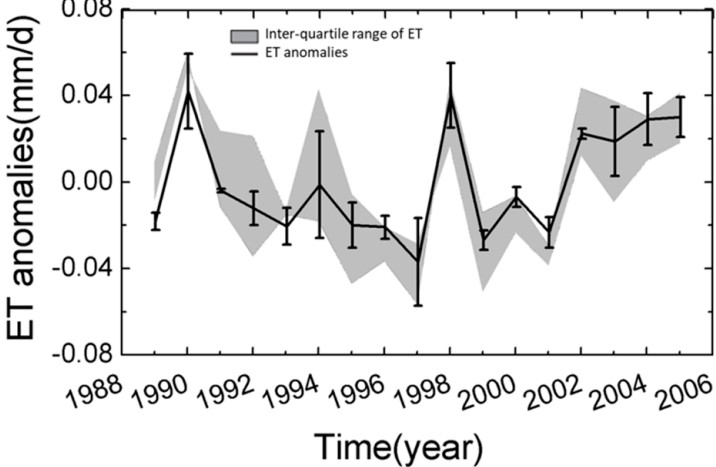

**Figure 6.** Evolution of national mean ET anomalies from 1989 to 2005 in China. The black lines with error bars represent the annual ET anomalies, and the gray shadow is the quartile range.

Figure 7 shows the average ET trends of 1989–1998 and 1998–2005 in China and the two regions. The general trends nationwide and in both regions were negative before 1998, and all became positive after 1998. Whether before or after 1998, the trend in the energy-control region was always more significant than that in the water-control region. The trends of ET changes in China before and after 1998 have always been dominated by ET changes in the energy-control zone. Combining this information with the trend distributions for winter and summer in Figure 3 suggests that the significant increase in national ET after 1998 was largely due to a significant increase in ET in the energy-control zone during the summer.

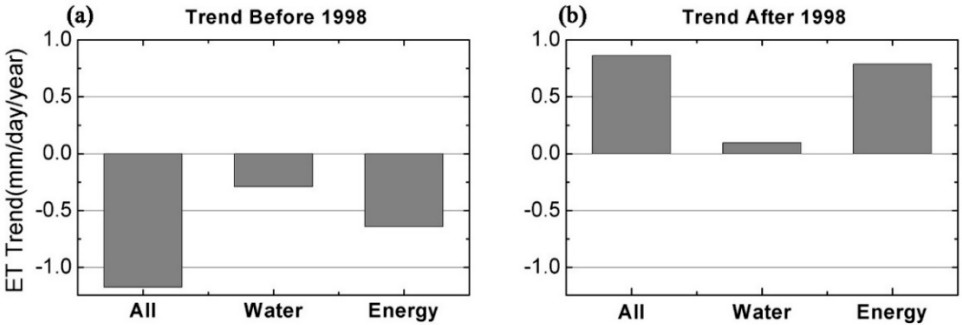

**Figure 7.** The average ET trends of 1989–1998 (**a**) and 1998–2005 (**b**) in China and the two divided regions. "All" represents the general trend of ET in China, "Water" represents the water-control region, and "Energy" represents the energy-control region.

The evolution of ET in the two areas with different restrictions is further shown in Figure 8. In the water-control region, the interdecadal changes in ET around 1998 are not obvious, and the summer range is large but does not affect the annual average. In the energy-control region, there are significant interdecadal changes. ET decreased before 1998 and increased significantly after 1998, which is consistent with the national average ET evolution (Figure 6). Overall, the significant interdecadal changes in ET in China around 1998 are mainly dominated by the interdecadal changes in ET in the southern energy-control region. That is, PET is the cause of interdecadal changes in ET. The increase in ET after 1998 is mainly caused by the significant increase in ET in the southern demand-restricted region during summer.

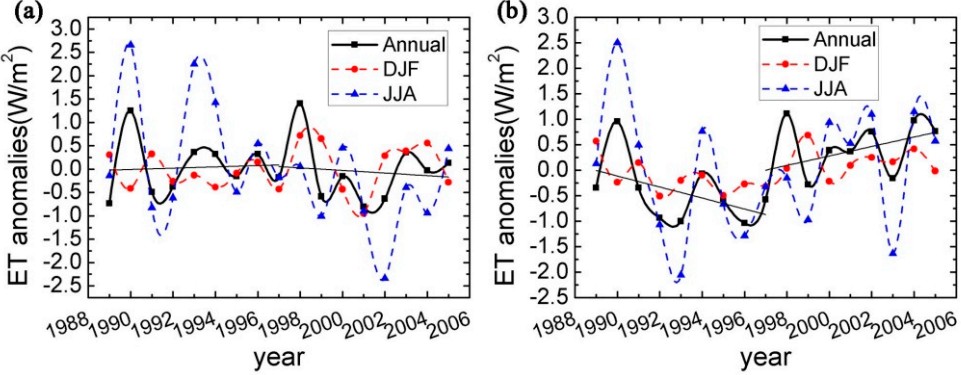

**Figure 8.** The evolution of ET anomalies in the water-control (**a**) and energy-control (**b**) regions. The black solid lines represent the annual average anomalies, the red dotted lines show the winter averages, and the blue dotted lines indicate the summer averages. The black line segments are the lines fitted to the ET anomalies before and after 1998.

### 3.4. Dominant Modes of Evapotranspiration in the Energy-Control Region

The EOF decomposition of evapotranspiration in the energy-control region of South China shows that the fractional variances explained by the first three modes account for 57.3%, 13.5%, and 7.4%. The variances in the remaining modes decrease rapidly after that, and the two leading modes are mainly analyzed here. The first mode accounts for 57.3% of the total variance and can be considered the main reason for ET changes. Such high explanatory variance can be obtained not only by avoiding the uncertainty in the reanalysis data used in previous studies, but also by appropriate region selection. The first mode only accounts for 20% in Su et al. [33] because they used 5 reanalysis datasets and applied the EOF to the whole country. As shown in Figure 9b, the time series of the principal component of the first mode (PC1) is almost all positive before 1998 and becomes negative after 1998. That is, the first mode represents the major spatial variability in ET, while the corresponding PC1 represents the decadal variability in ET. Combining these data with the spatial distribution in Figure 9a, the first mode of ET is negative in all of the energy-control region, indicating that ET decreases before 1998 and increases after 1998, which is consistent with the results of the above analysis.

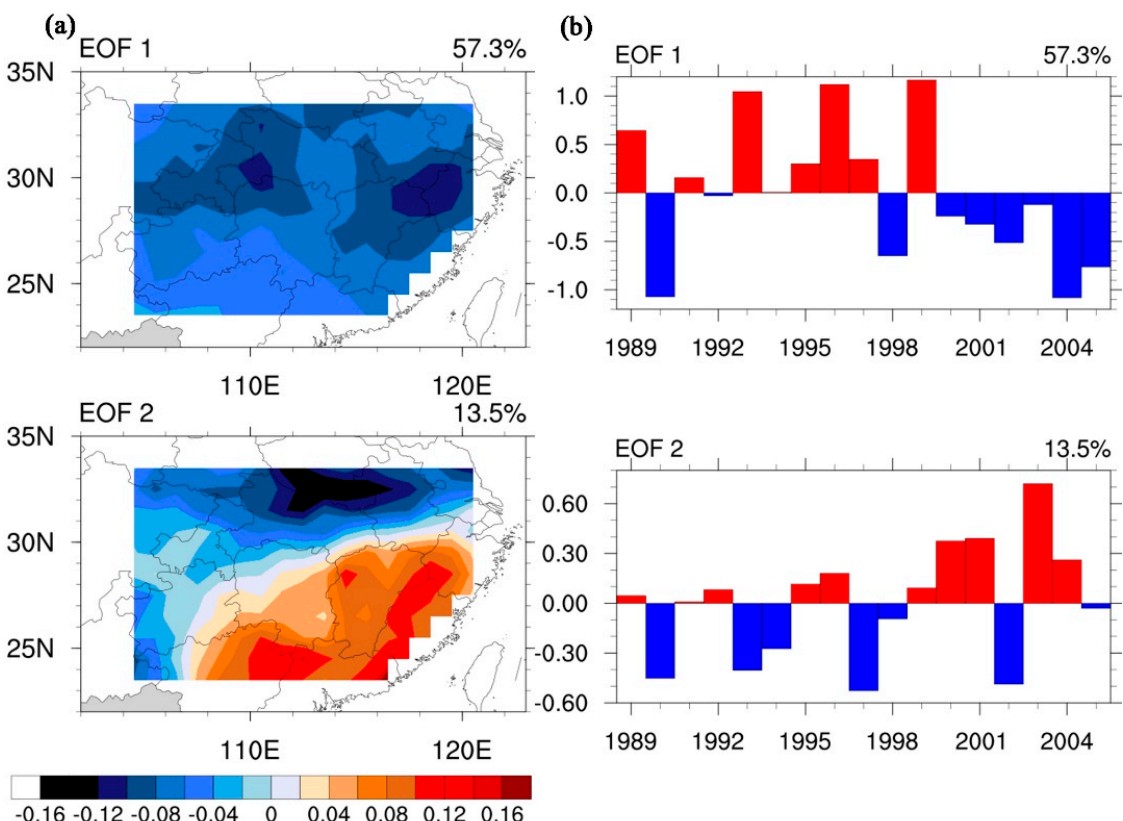

**Figure 9.** The spatial distribution (**a**) and principal components (**b**) of the first two modes in the energy-control region. The numbers in the upper right corner of the figure represent the explanatory variances in the first modes.

The time coefficient change in the second mode (PC2) implies that this mode represents the interannual change in ET in the energy-control region, and the interannual change in ET has an oscillation period of approximately four years. However, notably, the explanatory variance of the second mode is only 13.5%; i.e., the interannual variation is not obvious, and ET is mainly manifested as significant interdecadal variations in the energy-control region of South China. These results are consistent with the analysis for the merged mean ET in NCEP2 and MERRA2 (Figure S2). The fractional variances explained by the first two modes account for 47.6% and 13.8%, respectively. The time series of

the principal component of the first mode (PC1) represents the decadal variability, and PC2 represents the interannual variation.

Since evapotranspiration in the energy-control region is dominated by potential evaporation, the contributions of temperature, net radiation, relative humidity, and wind speed that affect potential evaporation to evapotranspiration are further analyzed here. Figure 10 shows the evolution of PC1 compared with temperature, net radiation, relative humidity, and wind speed from 1989 to 2005 in the energy-control region. The evolution of temperature and net radiation is similar to that of PC1 before 1998, both of which show a year-by-year positive increasing trend. However, the net radiation continued to increase after 1998, and only the temperature began to decrease year by year, as in PC1. There are no obvious interdecadal variations in the relative humidity and wind speed, and the changes in wind speed are mainly reflected in the interannual variation. PC1 and PC2 are correlated with temperature, net radiation, relative humidity, and wind speed. PC1 has the highest correlation coefficient with temperature, which is 0.326 and passes the 99% significance test. Since the variance of the second mode is very small, PC2 is poorly correlated with all meteorological factors, and the correlation coefficient with wind speed is the highest. That is, the interdecadal change in the energy-control region is mainly affected by climate warming, and the weak interannual change is mainly related to the annual change in wind speed in South China.

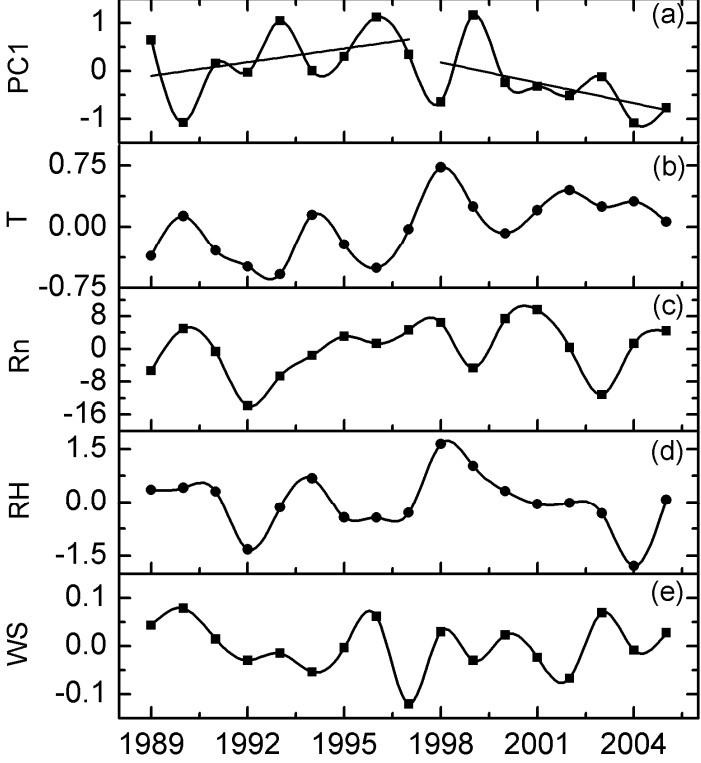

**Figure 10.** The evolution of PC1 (**a**) and temperature (**b**), net radiation (**c**), relative humidity (**d**), and wind speed (**e**) from 1989 to 2005 in the energy-control region. The black line segments are the lines fitted to the ET anomalies before and after 1998.

*3.5. Dominant Modes of Evapotranspiration in the Water-Control Region*

Similarly, the EOF is applied to evapotranspiration in the water-control region in North China, and the variances explained by the first three modes are 33.8%, 23.0%, and 12.8%. The two leading modes are mainly analyzed here, accounting for 56.8% of the total variance. Unlike the energy-control region, the water-control region has explanatory variances between modes that do not decrease as rapidly. As shown in Figure 11b, the time series of the principal component of the first mode (PC1)

represents the interannual variability in ET. Combining this information with the spatial distribution in Figure 11a, the first mode of ET is negative in all the water-control regions, indicating that the interannual variation in ET is the opposite of PC1.

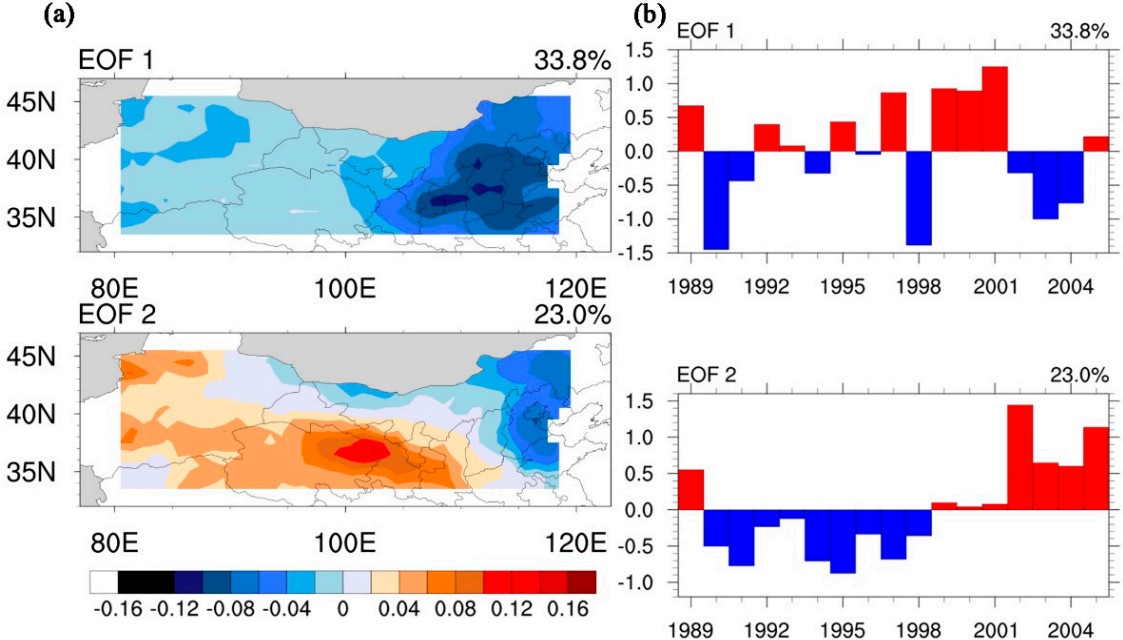

**Figure 11.** The spatial distributions (**a**) and principal components (**b**) of the first two modes in the water-control region. The numbers in the upper right corner of the figure represent the explanatory variance in the first modes.

The time series of the principal component of the second mode (PC2) is almost all negative before 1998 and turns positive after 1998. That is, PC2 represents the interdecadal variability in ET. However, notably, the explanatory variance in the second mode is small, and in contrast to the energy-control region, the water-control region of North China has ET that is mainly manifested as an interannual variation. The spatial distribution of the first two modes is similar to the EOF decomposition for merged mean ET in NCEP2 and MERRA2 (Figure S3). The major differences are that the time series of the principal component of the first mode (PC1) represents the interdecadal variability in ET for reanalyses, while the time is extended to 2016.

Correlation analyses between the time coefficients of the two leading modes in the water-control region and each meteorological factor are performed. PC1 has the highest correlation coefficient with precipitation, which is 0.59 and passes the 99% significance test. Since the variance in the second mode is very small, PC2 has poor correlations with all meteorological factors. The water-control region is mainly manifested as the interannual change controlled by precipitation.

By carrying out a principal component analysis after dividing China into climate zones by judging whether local precipitation can meet the demand of atmospheric evaporation, the interpretation of the variance of the first few main modes can be effectively improved to better capture the temporal and spatial variations in evapotranspiration. Previous studies have also noted that the evolution of evapotranspiration in China is mainly interdecadal [8,33]. A comparison of the evolutionary characteristics of arid and humid areas leads to the conclusion that the spatial and temporal variations in evapotranspiration in China are more clearly manifested by the interdecadal variation controlled by the temperature increase in the energy-control region and the interannual change dominated by precipitation in the water-control region. That is, global warming will continuously increase the actual evapotranspiration in humid regions, while there will be only annual oscillations with small amplitudes in arid regions. Further, we discuss whether the evolutionary forms of evapotranspiration

across the globe share the same characteristics and whether the evolution of ET will aggravate the climate differences in various climatic regions.

### 3.6. The Difference in Surface Energy Partition

Due to the different evolutionary forms of evapotranspiration in the energy-control region and the water-control region, the continuous difference under global warming will inevitably affect the surface energy partition and atmospheric water balance in the two regions. The partition of energy between sensible heat and latent heat, i.e., the Bowen ratio, can have a profound impact on the ecological climate and hydrological cycle at both regional and global scales [62]. The change in latent heat (i.e., ET) must be accompanied by the redistribution of surface energy, which will further significantly affect the boundary layer structure and its development. Hence, we analyze the variations in surface energy distribution terms from dry and wet land surface conditions. The sensible heat has different trends in the two regions (figure not shown), which means that different energy partition trends occur within the two regions. Here, the evolution of the Bowen rate in both regions and the different wetness degrees are shown in Figure 12. The reanalysis data of sensible heat and latent heat come from ERA-Interim.

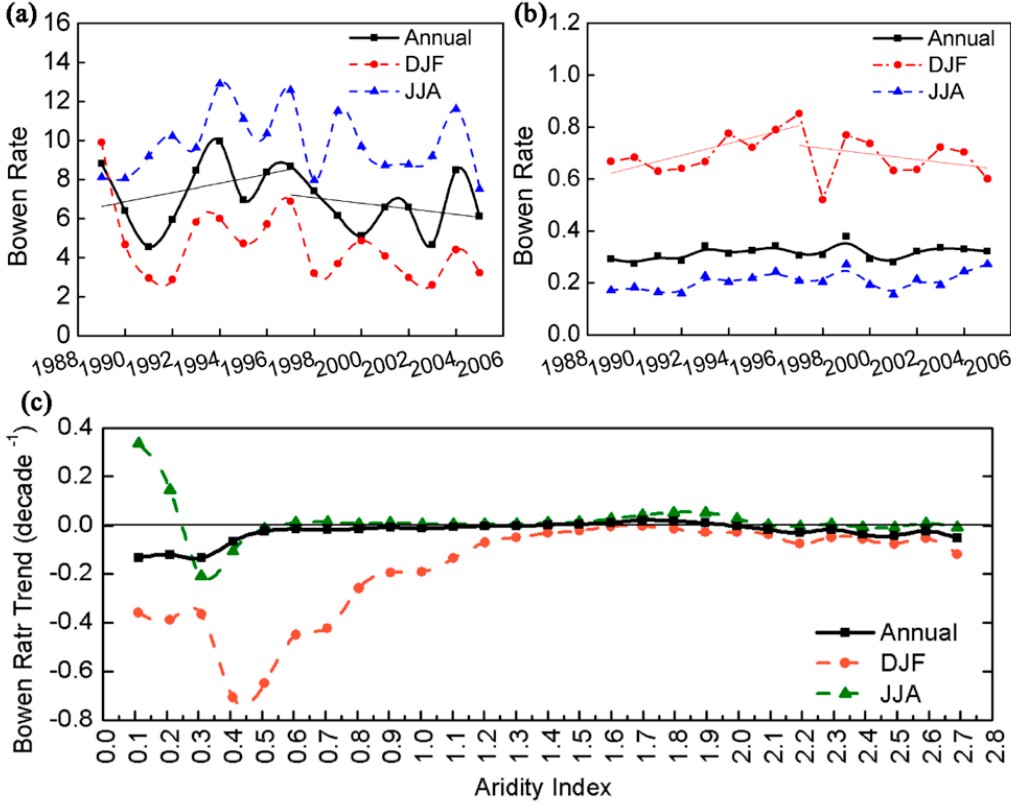

**Figure 12.** The evolution of the annual Bowen rate in the water-control (**a**) and energy-control (**b**) regions. The annual Bowen rate MK trends as a function of the AI during 1989–2005 (**c**). The black solid lines represent the annual average anomalies, the red dotted lines show the winter averages, and the blue dotted lines indicate the summer averages. The black line segments are the lines fitted to the ET anomalies before and after 1998.

In the water-control region (Figure 12a), the annual Bowen rate is much larger than 1, with sensible heat an order of magnitude larger than latent heat in summer. The BR first increases and then decreases in the water-control region, and the BR evolution shifts around 1998, whereas ET has no obvious change (Figure 8a). The BR diminishes rapidly in the winter after 1998 because precipitation in arid zones can have a significant effect on both sensible heat and latent heat. The BR is less than 1 in the energy-control region (Figure 12b), reflecting the large difference in energy partitioning between the

two regions. However, there is no clear change in the BR around 1998, and the annual BR has a small increasing trend in the energy-control region. Nevertheless, the BR is also decreasing in winter as in the water-control region.

Figure 12c shows the average BR MK trends as a function of the AI during 1989–2005. When the AI is greater than 0.5, there is no significant change in the annual average BR. The decreasing trend in BR occurs mainly in arid zones, as shown in Figure 12a. Koster et al. demonstrated that semiarid regions exhibit some of the strongest couplings between land and atmosphere [19]. Huang et al. also showed an enhanced warming effect in arid and semiarid regions [63]. According to Feng and Fu, 2013, arid and semiarid areas are defined when AI <0.2 and 0.2≤ AI <0.5, respectively [48]. The BR in arid areas has opposite trends in winter and summer, and a significant increase in the BR in summer can make arid areas dry and hot, which is an important cause of summer heat wave events [7]. While the BR in semiarid regions has a decreasing tendency in winter and summer, indicating a gradual shift in energy distribution, wetting of the Chinese semiarid zone has also been reported recently [35,64]. This decreasing trend in the BR is almost always contributed by winter across all AI ranges. That is, the BR changes accompanied by ET do not increase climatic differences between energy- and water-limited zones and will optimize energy distribution in semiarid zones, but increases in the BR in arid summers can significantly affect local weather.

## 4. Conclusions

Evapotranspiration, as a key indicator of the land–atmosphere water cycle and the energy cycle, is of great significance to understanding and addressing the uneven distribution of water resources under global warming. China lies in the climate-sensitive East Asian monsoon region, and the scarcity of flux observation sites in China limits research on the large-scale evolution mechanism of ET. Here, datasets that combine multiple reanalysis data retrieved from satellites and global flux sites are used to discuss the evolutionary mechanisms of different climate regions in China as well as the regional climate feedbacks of evapotranspiration.

The applicability of the LandFlux-EVAL dataset in China is verified through a data comparison with FLUXNET stations in China. The distribution and evolution of the average evapotranspiration in China from 1989 to 2005 are obtained. On average, the ET value is the largest on the southeastern coast, reaching 4.0 mm/day, and from the southeastern coast to northwestern Xinjiang Province, ET shows a gradual decrease. ET increases in most parts of the country except for the northeastern part of Inner Mongolia and the part adjacent to the Tibetan Plateau and Sichuan Basin from 1989–2005. The national annual ET increases on average by 0.665 mm/day/year. There are significant interdecadal changes in the national average ET. ET decreases until 1998 and then increases significantly after 1998, which is the opposite of global ET evolution. From a comparison of the trends of ET in winter and summer, the increase in ET after 1998 is mainly contributed by the significant increase in ET in the areas south and northwest of the Yangtze River during summer.

To analyze the factors that influence actual evapotranspiration and potential evaporation under climate change, China is divided into two regions: one region is limited by water supply, and the other region is limited by atmospheric evaporation demand. Most of Xinjiang and Gansu Provinces and the area north of the Yellow River are restricted by soil moisture, while Sichuan Province and the area south of the Yangtze River are restricted by atmospheric evaporation demand. Combining this information with the winter and summer distributions of ET trends and the ET evolution in the two regions, there is no significant interdecadal ET change in the area restricted by soil moisture around 1998. Regardless of winter or summer, the evolution of ET is dominated by the change in ET in the southern energy-control region. After 1998, the increasing trend of ET in China is mainly caused by the increase in ET in the energy-control region in summer. That is, changes in PET against the background of climate change are the main reason for the interdecadal changes in evapotranspiration in China.

Furthermore, EOFs are applied in the regions with different restrictions, and the evolutionary forms of evapotranspiration in the two regions are quite different. In the energy-control region, the explained

variance of the first principal component of ET reaches 57.3%, and PC1 is mainly manifested as an obvious interdecadal change, which is controlled by global warming. In the water-control region, ET is mainly manifested as interannual variation controlled by precipitation. Such different feedbacks of evapotranspiration with respect to climate change in the two regions are also reflected in the difference in surface energy partition. The Bowen rate in the water-control region decreases, but the large partition differences between summer sensible heat and latent heat in the arid zone are further increased.

The above findings imply that the evolution of water cycle components varies greatly in different climatic regions. In arid and semiarid regions, which have strong land–atmosphere interactions, research needs to focus on the climate feedbacks of their water cycles. The results will provide information for water resource planning and management in China and will supply important background information for climate change studies at a regional scale. The influence of increased water vapor from local evaporation on heavy precipitation can be further analyzed in the future.

**Supplementary Materials:** The following are available online at http://www.mdpi.com/2073-4441/12/11/3250/s1, Figure S1: Evolution of national mean ET anomalies from 1989 to 2019 in China for LandFlux-EVAL, NCEP2, MERRA2 and ERA5, Figure S2: The spatial distribution (left) and principal components (right) of the first two modes in the energy control region for reanalyses, Figure S3: The spatial distribution (left) and principal components (right) of the first two modes in the water control region for reanalyses.

**Author Contributions:** Conceptualization, J.L. and L.Z.; Data curation, Q.G. and Z.Z.; Funding acquisition, L.Z.; Methodology, Q.G. and X.C.; Writing—review & editing, Q.G. All authors have read and agreed to the published version of the manuscript.

**Funding:** This research was funded by the National Natural Science Foundation of China grant number 41605005, 41627807 and 41630421.

**Acknowledgments:** This research is supported by the National Natural Science Foundation of China (grants 41605005, 41627807 and 41630421). The authors would like to thank the LandFlux-EVAL project, which is coordinated under the Global Energy and Water Exchanges (GEWEX), for providing global merged benchmark synthesis products which are available on the internet (www.iac.ethz.ch/url/LandFlux-EVAL). And the European Centre for Medium-Range Weather Forecasts (ECMWF) and National Centers for Environmental Prediction-Department of Energy (NCEP/DOE) for providing the reanalysis datasets used in this paper. We also acknowledge FLUXNET for the observational data (https://fluxnet.fluxdata.org).

**Conflicts of Interest:** The authors declare no conflict of interest.

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
