# Peer review of "Spatiotemporal Evolution of Evapotranspiration in China after 1998"

_water, doi:10.3390/w12113250_

Round 1
Reviewer 1 Report
The paper deals with the topic of the estimation of spatiotemporal evolution of Evapotranspiration and Potential evapotranspiration in China from 1998 to 2005 in the framework of climate change. The analysis is performed using water control and energy control regions to discuss the influential factors of ET change and this results in different spatiotemporal feedbacks to be investigated.
In general, the connection between the theory, data and results should be improved at some extends so that the paper become clearer and more informative. The references are proper and cited in the text appropriately; the background literature and study rationale are quite clearly articulated but the questions presented at the beginning of the paper are partially answered.
The subject of the paper is suitable for the journal and the authors properly highlighted, especially in the Introduction, those elements in the article to be considered innovative for the topic, showing the confidence of the authors in the state of the art; however, the key aspects of the paper are only partially answered and the paper does not provided all the necessary information to follow the performed analysis: the paper still needs to be improved to guarantee that each step of the presented processes can be reproduced. Furthermore, the authors do not describe the limitations of the proposed approach, how they face and justify these limitations and those elements which could be further investigated in the future.
The organization of the article and the way by which the information and the key messages are conveyed are proper but should be improved especially the Data and Methods sections. The paper is well-written, tables or graphics are clear to read, labelled appropriately, informative and each table or figure is reported in the text.
As a general note, I would encourage the authors to provide more quantitative results instead of qualitative evaluations. An example can be found at line 118 when saying the data are “highly reliable”: while it is true that a reference is reported I would encourage the authors to provide in the text a quantitative estimate on how much the data are reliable (possibly with an associated error) and on which criterion this evaluation is based. Other examples can be found in the text. Finally, I would highlight that little investigation on the involed uncertainty affecting the analysis is reported (see figure 3 for confidence bands of the ET anomalies as a reference in whole paper): further analyses are, in my opinion, needed to be associated to the results.
In the following, some suggestions and ideas are reported which could be useful to make the paper more exhaustive and easier to be followed. The aforementioned analysis and particularly the need of few clarifications of some aspects of the paper make the manuscript difficult to fully assess at this time and I therefore recommend major revision.
Abstract and Introduction section
Both the Abstract and the Introduction provide a clear picture of the background of the study; it is well-structured, informative and up to date.
Study area and Data
No information is provided regarding the study area which should be briefly introduced. Datasets applied in the study are clearly listed: however, a more detailed description of the data is necessary both to understand better the paper and to make the simulations to be reproduced. It would be interesting if the authors could provide some plots of the data and some statistics indices to describe them. Moreover, I suggest that lines from 168 to 210 to be considered part of the Data section instead of the results sections.
Methods
I would personally improve the section with a more detailed analysis of the models and on the relative equations. In general, more in depth information about the methods and techniques used is needed; as previously mentioned, while it is true that some references are reported, the main equations should be reported in the text to help the reader following the results and discussion sections. Each variable in the equations should be described together with the relative measurement unit. For the detailed description of the methods, I would encourage the authors to use an annex, added to the text, for e clear explanation of the method applied.
Results and discussion
My suggestion is to report the part from line 227 to line 264 before the Results and Discussion session, which starts with the paragraph from line 214 to line 226 (“Preliminary trends results of the whole study area”).
The description of the regional subdivision is clear and starting from line 268, the results discussion is quite clearly articulated and presented.
Line 269: 1998 turn out to be a milestone during the analyses of the results; I would devote some part of the paper to explain why this year is a milestone which is not very clear at the present state of the article
In general, the plots are meaningful and the analysis of the results proper and logically structured. As previously said, I would encourage the authors to explain more critically the reported results linking them to the state of the art: indeed, a good number of proper references are reported in the section; however a comparison of the obtained results is necessary to improve the paper which, at the present state, does not provide a validation of the obtained results nor a confidence analysis on them: indeed, this could be a way to assess quantitatively the performance of the results enhancing the value of the discussion.
Reviewer 2 Report
The study period 1989-2005 does not give the authors right to draw such categorical conclusions as they did. The research material is too short to talk about trends before and after the year 1998. This could be only a prelude to further research, which points to some emerging tendencies that may develop in the future. The authors cite climate change and conclude their research in 2005! From today's perspective, it is already archival material. Probably, in 2007, this article would have been interesting and contributed a lot to the consideration of climate change and its consequences/ impacts, but not today. Over the next 15 years, to this day, the changes have gone further and it would be interesting today to check if the tendencies that the authors found are still going on or the process is intensifying or remaining at the same level, or maybe it was another climate fluctuation.
So, despite quite accurate selection of research methods and a good study, I cannot recommend the paper for publication. The only option I see for this material is to update the research series to 2019 and re-write the paper. Therefore, I will no longer discuss specific paragraphs of the text as its wording seemed to be missed or questioned to me.
Reviewer 3 Report
The authors have conducted a detailed study and the manuscript is well written. A minor comment I have is:
Figure 5: The colour appears at different contrasts. Provide colour scale also.
Round 2
Reviewer 1 Report
The authors answered the reviewers’ suggestions and questions: they followed some of them while, in the other cases, provided explanations to the comments and proper references supporting their positions. In general, the authors changed the manuscript which has been improved. Some parts of the paper have been re-organized; they clarified some aspect and provided more details and information to reproduce the results which are mentioned or reported. The text is well written. My very final note is to encourage the authors to highlight future developments of the paper and in general to the paper topics, which could remain a limitation of the study if left unexplained. In conclusion, I think this paper can be a contribution to the topic and to the journal.
Reviewer 2 Report
I am afraid that an explanation from cover letter does not work for me. If 1998 was a vital moment, nothing prevents the authors from extending the data series to the present, and after analyzing it, sticking to their conclusions, if they are still valid.
Summing up, I still think that the authors do not have the right to draw such categorical conclusions as they did and that the material should be updated to the present to be valuable and interesting for today's reader. I stick to my original opinion and do not recommend the paper for publication.
Round 3
Reviewer 2 Report
Once again, the authors tried to justify the applied methodology and the conclusions that they drew from the research. First of all, they finally explained why their research period ends in 2005. They downloaded other available re-analyses to provide additional information on the variability of ET after 2005 and shared them with the reviewer in a cover letter. In the context of the conducted research, this information is very important. All details that have been delivered to the reviewer should also be available to the journal reader.
The authors added a supplement to the text in which they placed 3 figures. It is a very good move, but it is not enough in my opinion. First of all, there is no reference to the supplement in the main text. The reader would not know about the existence of additional material that could facilitate perception of the text or dispel doubts. I also believe that at least some excerpts from the description provided in the cover letter to the reviewer should be placed in the appropriate places in the main text. Such places could be the methodology section for Fig s1 and the discussion for s2 and s3. Such a procedure will significantly affect the clarity of the text.
In addition, although the authors adhered to the original conclusions by adding the last paragraph, they indicated that the changes that occur in the environment / water cycle along with climate changes are not clear-cut and require further investigations/researches.
To sum up, I recommend the paper for publication but after revision.
